# Variation in Fungal Community in Grapevine (*Vitis vinifera*) Nursery Stock Depends on Nursery, Variety and Rootstock

**DOI:** 10.3390/jof8010047

**Published:** 2022-01-03

**Authors:** Sarah B. Lade, Dora Štraus, Jonàs Oliva

**Affiliations:** 1Forest Science and Technology Centre of Catalonia (CTFC), 25280 Solsona, Spain; 2Joint Research Unit CTFC-AGROTECNIO, 25198 Lleida, Spain; dora.straus@udl.cat (D.Š.); jonas.oliva@udl.cat (J.O.); 3Department of Crop and Forest Sciences, University of Lleida, 25198 Lleida, Spain

**Keywords:** metabarcoding, *Vitis vinifera*, grapevine trunk diseases (GTDs)

## Abstract

Grapevine trunk diseases (GTDs) are caused by cryptic complexes of fungal pathogens and have become a growing problem for new grapevine (*Vitis vinifera*) plantations. We studied the role of the nursery, variety, and rootstock in the composition of the fungal communities in root collars and graft unions of planting material in Catalonia (NE Spain). We compared necrosis and fungal communities in graft unions and root collars at harvest, and then after three months of cold storage. We evaluated combinations of eleven red and five white varieties with four common rootstocks coming from six nurseries. Fungal communities were characterized by isolation and metabarcoding of the ITS2 region. Our data suggests that nursery followed by rootstock and variety had significant effects on necrosis and fungal community structure in graft and root tissues. Within the plant, we found large differences in terms fungal community distribution between graft and root tissues. Graft unions housed a significantly higher relative abundance of GTD-related Operational Taxonomic Units (OTUs) than root collars. More severe necrosis was correlated with a lower relative abundance of GTD-related OTUs based on isolation and metabarcoding analyses. Our results suggest that nurseries and therefore their plant production practices play a major role in determining the fungal and GTD-related fungal community in grapevine plants sold for planting. GTD variation across rootstocks and varieties could be explored as a venue for minimizing pathogen load in young plantations.

## 1. Introduction

Grapevine trunk diseases (GTDs) attack the wood of grapevines (*Vitis vinifera*) and devastate vineyards worldwide. They have been recognized as a problem for well over a century, but their impact has increased significantly in recent decades [1]. There are several reasons for this change, one being the sudden intensification of grapevine planting and large-scale production in the early 1990s, which increased the global movement of potentially contaminated material [2]. Methodologies used in the wine industry modernized, and operations were often transferred from family-sized businesses to large-scale productions that were held to a higher standard of regulation. Such changes did not, however, take place at the nursery scale, leaving guidelines governing the quality of the planting material as they were [3].

As a result, vine defects originating in nurseries have become a recurring problem. The issues are most commonly attributed to the cutting quality, nursery practices and cold storage conditions [2,4,5]. These defects are then exacerbated in the field, especially as the increased demand for higher production puts additional stress on young vines by incorporating practices that favor fungal infection, such as training vines in high density spur-pruned trellises that are mechanically pruned, or leaving plants with more wounds [1]. The phasing out of the chemical products effective against GTD fungus has also complicated control efforts. Sodium arsenite, benzimidasole fungicides and methyl bromide were deemed toxic to both humans and the environment in many countries in the early 2000s and the discovery of active ingredients or biocontrol agents that could serve as a replacement option has been ongoing [6,7,8]. Consequently, in-depth study of the grapevine mycobiome and its interaction with the plant has been cited as necessary to fully understand the complexities of GTDs [9], and for technological development in the control of GTDs [10].

Grapevine trunk diseases are characterized as colonizing the wood of the perennial organs in the vines, causing wood necrosis and discoloration, vascular infections and/or white rot [8,11,12]. The external symptoms include leaf stripe, apoplexy of some or all of the plant, black measles on berries, and stunted shoots, which can appear suddenly or over the course of several years, and may or may not overlap with the internal decay [8]. Grapevine trunk disease pathogens are phylogenetically unrelated, belonging to different families, orders and even phyla, although infected plants may reveal similar symptomatology [9]. Usually, exact symptoms and the timing of their appearance varies among the diseases, as some of the fungi can live for years in the wood without causing any symptoms at all. The fungi that cause GTD symptoms to appear sooner are Petri disease (*Phaeomoniella chlamydospora, Phaeoacremonium* spp., *Cadophora luteo-olivacea, Pleurostoma richardsiae, Cephalosporium* spp. and *Acremonium* spp.) [2,13] and Black-foot disease (*Campylocarpon, Cylindrocladiella, Dactylonectria, Ilyonectria* and *Neonectria* spp.) [2]. Petri- and Black-foot diseases are more prone to attack young and replacement vineyards as symptoms will often appear several years after vines are planted in the ground when winter pruning begins (at three years), to when they reach maturity (at five years). In contrast, GTDs that affect mature vineyards are known as the Esca complex (*P. chlamydospora, Phaeoacremonium* spp., *Fomitiporia* spp. and *Stereum hirsutum*) [14,15], Botryosphaeria dieback (*Diplodia seriata, Botryosphaeria dothidea, Lasiodiplodia theobromae and Neofusicoccum parvum*) [16,17], Phomopsis dieback (*Phomopsis viticola*) [18] and Eutypa dieback (*Eutypa lata* and *Diatrypaceae* spp.) [19]. These GTDs are more likely to surface when the plants near 10 years of age, and any time after.

In Catalonia (NE Spain), the increased occurrence of sudden losses incurred by GTDs in 3–5-year-old vines poses a serious threat to the current economy and future outlook of the region. There are 10,000 wine growers and 853 companies with 11 denominations of origin (D.O.) [20]. The regional industry generates a turnover of 1.18 billion euros annually and represents 19.9% of the wine sector in Spain, making it a growing international marketplace [20]. A total of 55,118 hectares (ha) are currently utilized for vineyards, and more land has been approved for plantations in the coming years [21]. Consequently, this region has become a desirable place for young growers to settle and invest. However, with the estimated cost of establishment at 14,860 euros per ha over the 3 years required to plant and install all infrastructure needed for a successful operation [22], it is paramount that vineyards are equipped with healthy plant material, and the proper knowledge and tools to prevent GTD losses.

Growers lack information on the likelihood of finding GTDs depending on commercial factors such as nursery, scion, or rootstock variety. There are not any cultivars or species in the genus *Vitis* that have expressed complete resistance to GTDs [11], though some have differed in their sensitivity to the development of GTD foliar symptoms [23], suggesting that degrees of varietal resistance might be under genetic control [24,25]. Work has also been done to understand the degree of sensitivity to the disease of the rootstock, finding that some rootstocks or crosses are more resistant to certain GTDs [26,27]. For Esca disease in particular, higher sensitivity was seen for grapevine grafted onto rootstock with a higher tannin content [28]. Regardless, the final disease or resistance results from an interaction among the virulence of the pathogen, the capabilities of the host-plant defense, and the environmental conditions [29].

The mechanism by which GTD-related fungi shift from a latent to a pathogenic state and trigger the expression of disease is still a matter of debate [1]. The main hypothesis is that young vines infected with the pioneer fungi *P. chlamydospora* or species of *Phaeoacremonium* can later develop symptoms following the colonization by other basidiomycetes, although Koch’s postulates have not yet been completed to support such claims [15]. It has also been hypothesized that biotic or abiotic stresses act as an inciting factor in the shift of fungi to a pathogenic state [30], but this hypothesis also lacks substantial supporting evidence. Therefore, thoroughly mapping grapevine fungal community structure and microbiome is the next logical step in clarifying community dynamics and understanding how GTDs evolve in various contexts. Thus far, DNA metabarcoding approaches have revealed a higher diversity of taxa and accurate relative abundances in samples coming from the vineyard [31,32,33], though culture-independent studies that used the NGS approach to learn more about the grapevine endosphere are scarce [33,34]. Only one study investigated the mycobiome of GTD-affected plants to understand the spatial distribution of the communities present in each plant, linking microbial communities of the wood and the expression of leaf symptoms of Esca [9]. Work has also been done to analyze the biogeographic characterization of the microbial community structure associated with grapevine via grape samples, demonstrating that both Chardonnay and Cabernet Sauvignon varieties, collected across the four major wine regions in California, USA, displayed a specific associated microbiome, suggesting a regional pattern [35]. Furthermore, the grape cultivar and year of production influenced this microbial structure [10].

In this study, our aim was to explore variation of the fungal community inhabiting grafted grapevine nursery-stock, with the objectives of investigating each possible source of this variation, and focus on those connected with higher rates of GTD-related fungi. We investigated (1) the role that nurseries were playing in the proliferation of GTD-related fungal communities prior to plant distribution, and tested if storage had a significant effect on the microbial communities; (2) if the presence of GTD-related fungi was correlated with specific varieties and rootstocks; and (3) whether there existed a link between trunk necrosis and the structure of GTD-related fungal communities.

## 2. Materials and Methods

### 2.1. Plant Material

The experiment was conducted on healthy-looking bench-grafted bare-root plants that had been propagated as one plant in the field for a year. Plants were collected from nurseries at two sampling times: once during the week of harvest from the field in late autumn/winter (“pre-storage”), and again in early spring (exactly three months later; “post-storage”), after plants had been bundled, bagged, and held in the nurseries’ own cold-storage facilities at 4 °C, subject to the conditions and practices of each individual nursery. In all cases, pre-storage plants received minimum handling and treatment by the nurseries, while post-storage specimens underwent typical preparation for distribution, in which roots were trimmed to 10 cm and graft unions were dipped in paraffin wax in the nursery workshop. Six nurseries from the Catalan regions of either Girona, Tarragona or Barcelona participated in the study.

The plant material included three biological replicates of each scion-rootstock combination pre- and post-storage. We collected an array of red and white wine varieties (16 total), which depending upon the production of each nursery, grafted onto one the four most commonly-employed rootstocks in the region of Catalonia at this time: 110 Richter (R-110), 140 Ruggeri (RU-140), Selection Oppenheim 4 (SO4) and 41B (for white only). The 11 red varieties included: Autumn Royal, Cabernet Sauvignon, Caladoc, Carignan, Garnacha tinta, Merlot, Pinot Noir, Sumoll, Syrah, Tempranillo and Xarel.lo vermell; while the five white varieties collected were: Chardonnay, Macabeu, Malvasia, Parellada and Xarel.lo. The study was designed around color-rootstock combinations (with scion/varieties falling within the color category) because there were not enough replicates of each variety to cross them with each rootstock and nursery. Each color-rootstock combination was replicated at a minimum six times total, and from at least three nurseries (Table 1). The more common use of certain rootstocks over others (R-110 vs. 41B, for example) is reflective of its frequency of use in Catalonian nursery-stock.

### 2.2. Necrosis Evaluation

At the time of analysis, which was immediately after leaving the nurseries and being transported to our labs, each plant stem was cut into transversal sections so we could observe the necrosis present. The length (cm) of necrosis starting in the root collar and graft unions was measured as it extended along the inside of the stem. Specifically, root collar necrosis was measured from the base of the stem after all roots had been trimmed, and upwards; carefully quantifying the necrosis present on either side of the cambium and finally taking an average of the two measurements. Graft union necrosis was measured in a similar way, starting the measurement from where the graft could be seen meeting the rootstock. The morphology of the necrosis was also noted (discoloration, black spots, triangular formations).

### 2.3. Fungal Isolates

For all samples, isolates were gathered from wood chips that had been cut from regions next to cambium of the root collar and graft union. Fungal isolation was carried out by plating surface-sterilized samples cut into 0.5 cm long pieces on 0.3% malt extract agar (MEA) media following Santamaría et al. [36], which was amended with 0.10 g/L of chloramphenicol. Samples were incubated at 20 °C in the dark for ca. 2–3 days. A maximum of three morphologically distinct colonies per sample type (root collar or graft union) were transferred onto potato dextrose agar (PDA) media amended with 0.10 g/L chloramphenicol and stored at 20 °C in the dark for one week to provide active mycelial growth for DNA extractions. If morphological differences were not obvious, three isolates per shoot were taken at random. Mycelial growth was monitored to ensure the development of pure isolates, and further re-isolations were conducted as necessary.

Mycelium was scraped from the surface of the colony and placed in sterile Eppendorf tubes with 200 μL of NaOH (0.5 M). Samples were then vortexed briefly and centrifuged 12.0× *g* for 30 s. For the template, 5 μL of the supernatant was transferred to another tube containing 495 μL TrisHCl (0.1 M). PCRs were run by adding 4 μL template solution in 25 μL reactions with DreamTaq polymerase (Thermo Fisher Scientific, Waltham, Massachusetts), following manufacturer’s recommendations and using universal primers ITS1F (5′-3′: CTTGGTCATTTAGAGGAAGTAA) and ITS4 (5′-3′: TCCTCCGCTTATTGATATGC). Each sample was amplified using the following cycling conditions: an initial denaturation step at 95 °C for 3 min, 30 to 35 cycles at 95 °C for 30 s, 60 °C for 30 s, 72 °C for 1 min, and a final elongation at 72 °C for 6 min. Products were purified and sequenced by Macrogen Co. Ltd. (Seoul, South Korea). The sequences were cleaned and clustered using a 98.5% confidence threshold with DNASTAR software and blasted in GenBank, with a preference given to type material. The OTU identities were assigned according to match thresholds of >98% for species, >97% for genus, >95% for family, >92% for order, >90% for class and >80% for phylum. Sequences with <80% match that could not be identified as being fungal sequences were discarded from the dataset.

### 2.4. Library Preparation for Metabarcoding

For all samples, specimens were gathered at the same time that isolate wood was gathered, from wood chips that had been cut from regions next to cambium of the root collar and graft union. The wood chips were immediately frozen for later analysis. Surface-sterilized graft union and root collar samples were defrosted and chipped with sterile hand shears. Fifty mg of each sample was manually ground using liquid nitrogen and ca. 100 mg of sand. The pestle and mortars used for extraction were cleaned using a 5% NaOCl solution to avoid cross-contamination of DNA between samples. DNA extraction was conducted with some adjustments to the NucleoSpin© Plant II protocol by Macherey-Nagel (2018) from [37] to improve extractions from woody material. The ITS2 region was used as the metabarcoding marker, and amplifications used ITS4 and ITS7 tagged forward and reverse primers to enable de-multiplexing following [38]. PCR reactions were conducted at 57 °C in triplicates, and 32 cycles were used to obtain faint bands corresponding to the linear phase of the amplification. PCR products were pooled and cleaned using bead suspension and magnetic separator according to the NucleoMag^®^ NGS protocol. DNA concentrations were assessed with Qubit™ (Waltham, MA, USA) and the eight equimolar mixtures (pools), totaling 740 samples, were sequenced with 300 bp paired-end read lengths using two lanes of Illumina MiSeq at the Centre for Genomic Regulation (CRG; Barcelona, Spain).

### 2.5. Quality Control and Bioinformatic Analysis of Metabarcoding Data

The application of quality control, screening and clustering of sequences was conducted using a similarity threshold of 98.5% with the bioinformatics SCATA pipeline (scata.mykopat.slu.se). Sequences with less than 150 base pairs per read were discarded using the amplicon quality option. In addition, primer mismatches and tag jumps were identified and removed from the dataset. In total, 3,889,132 sequences passed quality control, and they were clustered into operational taxonomic units (OTUs) with a 1.5% similarity threshold. The dataset was further trimmed to ensure all OTUs occurred at least in three samples with >10 sequence reads each for the community analysis. We taxonomically classified each OTU by using the Protax software [39,40] in PlutoF (26 June 2021, https://plutof.ut.ee/), choosing a threshold value of 0.5 (plausible classification). This threshold is a stricter criterion than using a 97% sequence similarity threshold [41]. To exclude OTUs belonging to plants and non-fungal organisms, we performed a Least Common Ancestor (LCA) analysis (minimum score of 300 and a minimum identity of 90%) in MEGAN [42] with all OTUs that were not classified at phylum level by Protax. We kept the OTUs classified in MEGAN as ‘Fungi’, merged them with the OTUs previously classified by Protax, and classified them as ‘Fungi, unknown phylum’. Post-clustering curation of fungal OTUs was carried out with the ‘lulu’ package in R [43]. We used the same taxonomic assignment rules as for isolated fungi. The OTU table, ITS sequence of each OTU, and corresponding metadata are deposited in Figshare.

### 2.6. Statistical Analysis

#### 2.6.1. Necrosis

Statistical analyses and modeling were carried out using either R version 3.6.3 (64 bits) [44], or JMP^®^. The association between location of necrosis and plant characteristics was first examined by determining the significance of each characteristic in a one-way analysis of variance. The location of the necrotic tissue samples, either in the graft union (*n* = 369) or root collar (*n* = 370), was set as the response factor and two models were created. Sample (plant) characteristics included in the model as independent variables were: color, nursery, storage, rootstock (for root collar samples) and variety (for graft samples). In order to disentangle the interaction “variety x nursery” or “rootstock x nursery”, comparisons between varieties/rootstocks within nurseries and comparisons between nurseries within varieties/rootstocks were conducted. Post hoc comparisons were made using the Student’s t-test at the 0.05 level of significance; significant results were reported.

We correlated the relative abundance of GTD reads in each sample (calculated as a sum of GTD reads for each sample, over the total in each tissue type) with the length of necrosis present in the same sample.

To compare if particular GTD-related OTUs were associated with various degrees of necrosis in the two tissue types, we analyzed the first 100 most abundant operational taxonomic units (OTUs), comparing them as a percentage of the total number of reads for each tissue type via stepwise analysis. The top 100 most abundant OTUs were used for this analysis so as to remove rare species and have robust comparison with the more common species. Other parameters (plant characteristics; color, nursery, storage, rootstock, and variety) were also considered in the model, but locked as current estimates in JMP^®^.

#### 2.6.2. Mapping Fungal Communities

To know which factor(s) had caused significant shifts in fungal taxa, we conducted analysis of our metabarcoding data using R version 3.6.3 (64 bits) [44]. First, relative abundance of each OTU was standardized with a Hellinger transformation. The entire fungal community and then just GTD-related fungal community were tested by tissue types (graft union and root collar) using Permutational Multivariate Analysis of Variance (PERMANOVA; Adonis2 function) in the Vegan package (v. 2.5–5) [45]. The data set was then split by tissue type and analyzed for each factor (nursery, storage, variety, and rootstock). Variety was only analyzed for graft unions and rootstock only for root collars. Interactions were also investigated by running separate analysis for each of the levels of the factors. We used ‘vegdist’ function to calculate Bray-Curtis dissimilarities of the community matrices and tested for homogeneity of multivariate dispersion ‘betadisper’ (i.e., multivariate dispersion or beta diversity) using ‘permutest’ function. Changes in community composition were visualized with a principal coordinate analysis (PCoA) based on Bray distances amongst samples. More in-depth analysis was run on representative examples of the degree of variation between nurseries within varieties, and of the degree of variation of storage between nurseries influencing total fungal community and GTD-related fungal community composition. For this we examined variation in Macabeu plantings across nurseries, then the effect of storage in nurseries II and IV pre- and post-storage. The R^2^ and *p*-values shown in results are extracted from the PERMANOVA analyses performed with the adonis2 function.

The functions ‘diversity’ and ‘specnumber’ from the same vegan package were used to calculate alpha diversity indices, namely species richness, evenness, and the Simpson diversity index. To determine the efficacy of GTD occurrence rates based on reads vs. isolate data, we calculated the rates for each data set and compared them by various factors.

### 2.7. Indicator Species

An indicator species provides information on the overall condition of the ecosystem and of other species in that ecosystem. They reflect the quality and changes in environmental conditions as well as aspects of community composition. In this work, indicator species tests were used to compare frequency shifts of fungal taxa between tissue types for NGS data (OTU abundance). Indicator species tests were done with a multi-level pattern analysis in R with the ‘multipatt’ function of the ‘indicspecies’ package [46]. The ‘multipatt’ command results in lists of species that are associated with a particular group of samples and identify species that are statistically more abundant in combinations of categories. We searched for indicators by variety in graft unions and rootstock in root collar.

## 3. Results

### 3.1. Plant Attributes and Conditions Contributing to Necrosis

In both graft and root tissues, nursery had a large effect on necrosis length (Table 2). In graft unions, nursery explained 6% of the variation (*p* < 0.001) followed by variety that explained 1.9% of the variation (*p* = 0.017). Post-hoc analysis of the effects of varieties and color on graft necrosis revealed that red varieties were associated with significantly more necrosis than white ones (*t*-test, *p* = 0.049) (Figure 1A). Autumn Royal was the variety with the longest necrosis at graft level, and Carignan was the variety with the smallest necrosis. In the root collar, the rootstock RU-140 showed the shortest necrosis amongst the other rootstocks (Figure 1B).

### 3.2. Fungal Community Distribution

Clustering rendered a total of 732 operational taxonomic units (OTUs). Of these, 61 OTUs were identified to the genus level, 11 of which were considered as putative GTDs (Appendix A). The most abundant OTU, accounting for 33% of all reads was *Lophiostoma* sp., while the second most abundant was *C. luteo-olivaceae*, accounting for 15% of all reads (30.44% in graft unions and 4.61% in root collars) (Appendix A) [47,48,49,50,51,52,53,54,55,56,57,58,59,60,61,62,63,64,65,66,67,68,69,70,71,72,73,74,75,76,77,78,79,80,81,82,83,84,85,86,87,88,89,90,91,92,93,94,95,96,97,98,99,100,101,102,103,104,105,106,107,108,109,110,111,112,113,114,115,116,117,118,119,120,121,122,123,124,125,126,127,128,129,130,131,132,133,134,135,136,137,138].

The most significant variation in terms of both the fungal community and GTD-related fungi occurred between graft unions and root collars, accounting for 17% of the difference (*p* < 0.01) (Figure 2; Table 3A). The alpha diversity of each tissue type revealed that species richness was comparable between tissue types; however, when considering GTDs, the species richness was higher in graft unions than in the root collars. The Simpson Diversity Index indicated that root collars were comparatively more diverse than graft unions both considering GTD-related fungi, or the entire fungal community. Dividing the dataset by tissue type revealed that the most important factor contributing to the community variation in each tissue was nursery, accounting for 36% variation in graft unions, and 16% in root collars (Table 3A). A similar pattern was observed when only considering GTD related fungi (Table 3B). Variety was also an important factor, which accounted for 17% variation in graft unions and 13% in root collars (Table 3).

There was a positive correlation in the occurrence rates of GTD fungi between the two tissue types (Figure 3). Regardless, we found the number of GTD-related OTUs to be significantly different between the two tissue types (*p* < 0.001), with 48.6% of the OTUs identified as GTD-related fungus in graft unions, and only 7.2% in root collars (Figure 4A). No correlation between the relative abundance of GTD-related OTUs and necrosis was found. When considering the role of individual OTUs, we found more negative than positive associations between abundance and necrosis length (Figure 4B). The only identified OTU to be positively associated with necrosis in graft unions was *Cladosporium sp.*, which was negatively associated with necrosis in root collars. OTUs that were negatively associated with necrosis in graft unions were *C. luteo-olivaceae*, *Phaeoacremonium minimum*, *Alternaria sp.*, *Botrytis cinerea* and *Chaetomium* sp. Root collars had more species positively associated with necrosis, which were *Acrocalymma vagum*, *Chaetomium* sp., and *Preussia* sp. Species negatively associated with necrosis were *Lophiostoma* sp., *Cladosporium sp.* and *Fusarium oxysporum*. Finally, there was a significant, inverse correlation between graft union necrosis and the presence of GTD-related fungi; however, no significant association was found in the root collar (Figure 5).

### 3.3. Fungal Community Distribution within and between Nurseries and Varieties

In our study design, not all varieties/rootstocks were available in all nurseries (Table 1). Because of that, we analyzed the effect of nurseries and storage within varieties (Table 4). We found that the fungal and GTD-related fungal composition of plants of the same variety/rootstock could differ largely when delivered by one nursery or another. When we examined the effects of the most abundantly sampled variety, Macabeu, we observed that nursery explained ca. 60% and 45% of the variation for total and GTD-related communities, respectively (Figure 6A). For some nurseries like nursery IV, storage accounted for 30% of the variation, for others like nursery II, storage only explained 12% (Figure 6B). The variation explained between nurseries within varieties ranged from 43% to 65% for the total fungal communities, and from 22 to 44% for GTD communities (Table 4). The variation between rootstocks ranged from 18% to 44% for total communities and from 14 to 32% for GTD communities between nurseries. In contrast to nursery, the contribution of storage within variety was much smaller, and mostly non-significant in the case of GTDs.

Within nurseries, there was also variation across varieties, which explained from 15% to a 33% of the variation in fungal composition and from a 5% to a 33% of the variation in terms GTD-related fungal composition (Table 5). The effect of storage in the composition of the graft union was very dependent on the nursery. For some, it accounted for almost 45% of the variation in fungal composition (32% in terms of GTD-related fungi). Concerning the root collar, there was some variation across rootstocks within nursery, explaining from 6% to 24% of the variation in total fungal composition, and none for GTDs. The effect of storage was more important in this regard, explaining 3% to 28% of the variation, and 4% to 42% for GTDs (Table 5).

### 3.4. Relative Abundance of GTD Fungal Isolates and OTU Clusters

We cultivated and sequenced fungal isolates from the graft unions (n = 386) and root collars (n = 387) of the pre-storage plants in triplicates (taking three samples from each tissue in each plant). A total of 59 different species were identified, 15 of which were classified as GTD-related fungus or were of the same genera as previously identified GTDs. There was the same GTD-related isolate cultivation rate for the two tissues (GTDs-related fungi represented 18.28% of the total isolates in both cases) (Appendix A).

In both tissue types, nursery was the most significant factor considering the relative abundance of GTD isolates (ChiSq < 0.001 for both), while variety also had a significant effect in graft unions (Table 6).

The isolate species identified as GTDs differed from those that were identified in the metabarcoding analysis, except for *Diplodia* sp. and *Neofusiccoccum parvum*, which appeared in both analyses. Of all the GTD isolates, *Neofusicoccum* sp., *Botryosphaeria sp.* and *Diplodia* sp. were the most common taxa in graft unions, with relative abundance of 6.01%, 3.39% and 3.13%, respectively; while *Ilyonectria* sp., *Dactylonectria alcacerensis* and *Dactylonectria* sp. were most abundant in root collars (5.22%, 4.18% and 2.35%, respectively) (Appendix A).

### 3.5. Indicator Species Analysis

Indicator species analysis resulted in 31 OTUs significantly associated in graft unions and 39 in root collars (with an α ≤ 0.05; Appendix A). Of these, 22% were GTD-related fungi in graft unions, and 7% in root collars, a differential that was less than, but mostly consistent with the overall difference in relative abundance of GTD-related clusters in each tissue type. The variety with most association to GTD-related indicator species was Carignan (three of nine identified species were GTD-related fungi; *Diplodia* sp., *Phaeoacremonium* sp. and *P. chlamydospora*), followed by Malvasia (two of seven; both *Neofusicoccum* sp.). *Acremonium alternatum* was an indicator in Sumoll and *C. malorum* in Tempranillo. The only rootstock with significantly associated GTD-related OTUs was 41B (*Acremonium sp.* and *Truncatella* sp.). There was no overlap in GTD-related species amongst varieties nor rootstocks, iterating the specificity of these species for their host environments.

## 4. Discussion

Grapevine trunk diseases (GTDs) are caused by cryptic combinations of wood-inhabiting fungi. The intricacies of the relationships between these particular fungi, the plant hosts and environment are still not understood, so an in-depth study of the grapevine mycobiome has been cited as necessary to fully understand the complexities of GTD infections [9]. In this experiment, we used NGS technology to examine fungal communities in Spanish nursery-stock, and compare results with degrees of internal trunk necrosis present in above- and below-ground plant tissues.

The outcome of mapping the mycobiomes of two tissue locations in the stems of young plants highlighted the influence of nurseries in fungal community composition. Total and GTD-specific fungal communities were significantly different between graft and root tissues, however in both tissues, nursery was the factor explaining the largest amount of variation in both in terms of fungal amplicons or isolates. Our results suggest that either nursery practices or inherent biogeographic fungal community profiles unique to each nursery, play a crucial role in the development of GTDs in nursery stock. With only six nurseries it is not possible to disentangle such effects given the number of steps that could affect GTD variation before plants are sold to the grower. Previous studies have also cited nurseries and their practices as having an impact on vine health, referencing the variation in the many steps and practices involved in creating nursery-stock as a detriment to the quality of planting material [139]. Other research has directly demonstrated how infection by GTD pathogens occurs during the grafting processes in nurseries [2,140]. Certain grafting practices have proven to be more detrimental than others, for example the use of mechanical (omega) grafting machines, which allow plants to be grafted faster and with little training, but which have also been associated with a 50-fold increase in the incidence of esca in the field [141]. Regardless of the cause, our results show that nurseries explain 24% of the GTD variation in the grafts and 20% of the variation in the rootstock. Buying plants in nurseries with the best practices seems to be the best strategy to minimize the pathogen load and prevent future losses.

The abundance of GTD-related fungus decreased as necrosis increased. This trend was consistent for data from both analyses (isolate and NGS). Previous work has cited similarities in the community structure of symptomatic and non-symptomatic wood [9], but little explanation has been given as to why some communities induce necrosis, and others do not. There are a number of studies examining the principle factors involved in the proliferation and pathogenicity of GTDs, such as the different fungi present, how these fungi interact with the grapevine cultivar [11], physiological factors in the plant [142], and the environment [8]. The patterns observed at community levels were mirrored by correlations at species level. Both *Cadophora luteo-olivaceae*, and *Phaeoacremonium minimum* were negatively correlated with graft necrosis. *Cadophora luteo-olivacea* is a major pathogen in Petri disease, which is a form of Esca that effects young grapevines, and which in our case was associated with shorter necrosis. The lack of correlation between necrosis and GTD abundance does not downplay the fact that some plants more than others harbored a significant amount of GTDs which could trigger a disease later on.

Our study highlighted the importance of the methodology used to identify fungi: isolation and metabarcoding. These techniques rendered two distinct sets of GTD-related fungi in young nursery plants with little crossover: those that were determined from sequencing isolates collected from symptomatic tissues and those that could only be detected with NGS technology. From isolates, *Neofusicoccum* sp., *Botryosphaeria* sp., *Ilyonectria* sp., *Dactylonectria* sp. and *Diplodia* sp. were the most common GTD-related species, while *Acremonium alternatum*, *C. luteo-olivaceae*, and *P. minimum* were the most abundant NGS-detected species. Differences between isolation could stem from methodological biases due to growth on PDA media, or due to primer affinity. They could also reflect the viability of the organisms themselves. Before NGS-technology could be used to understand the entire mycobiome of symptomatic and asymptomatic plants, researchers like Hofstetter, Buyck [143] had found that when comparing asymptomatic and Esca-symptomatic plants, the incidence and abundance of Esca-related fungi were independent of the plant attributes, as all seemingly carried the same fungal parasitic load. These conclusions were drawn with full knowledge that “the term fungal community or mycota that were isolated represented only part of the culturable fungi and missed uncultivable fungal species”. Since there was no significant difference in the systematic structure of the mycota associated with asymptomatic and Esca-symptomatic plants, it was suggested that they are part of the normal mycota associated with adult grapevine plants [144,145].

Both variety and rootstock had an influence on necrosis and fungal community. In general, rootstock had a larger influence than variety, which may be explained by biological events tied to the plants’ capacity to store moisture from the soil or an earlier harvest time prior to grafting. More difficult to interpret are the mechanisms behind a higher prevalence of GTDs associated with one variety versus another. The fact that both nursery and variety have effects on necrosis and community structure may reflect the fact that nurseries take grafts from different mother plants, which can in turn harbor a different community of GTDs. We evaluated our results in context with recent findings by Chacón-Vozmediano et al. [146], to see if there were any varieties that were evidently more susceptible to developing necrosis across conditions. While there were not any direct correlations, our findings coincided in that varieties such as Malvasia, Tempranillo and Macabeu were more susceptible to developing GTD-related symptoms in the field tended to harbor more GTD-related fungi, while Pinot Noir and Parellada were less so (Appendix A). Our results showed Cabernet Sauvignon to be outwardly the least susceptible variety, while observations in the field showed it to rank quite high. The lack of correlation between field and nursery observations may also be explained by the large role played by the nursery producing the plants that were outplanted.

We detected the presence of several biocontrol agents as indicator species in rootstocks SO4 and 41B. *Alternaria* sp. was detected in SO4 rootstocks, and has been identified as a promising biocontrol agent against the *Vitis vinifera* pathogens *Plasmopara viticola*, the causal agent of downy mildew [147], and *Botrytis cinerea*, which causes Botrytis bunch rot [148]. The mode of action for *Alternaria* sp. is the production of resveratrol, an antioxidant compound known to increase resistance to stress and prolong the life of the organisms which it inhabits [149]. While there were not any GTD-related indicator species in SO4, this rootstock did have the most necrosis, so *Alternaria* sp. may be interacting with the rootstock to increase its tolerance to the stress of the necrosis, or may play a role in preventing pathogen colonization. *Lophiostoma* sp. is another potential biocontrol agent that was detected as as indicator species in 41B rootstocks. Specifically, *Lophiostoma* sp. was reported as producing antifungal compounds (in addition to metabolites that are effective against pathogenic bacteria), so has been cited as a potential biocontrol agent for vine-related pathogens, such as *B. cinerea* [56]. In our work, *Lophiostoma* sp. appeared accompanied by the GTD-related fungus *Truncatella* sp., a species that has repeatedly been associated with esca complex fungi [150], as well as *Acremonium* sp., which is a non-pathogenic fungus that has recently been found to be an indicator of symptoms (necrosis) in non-GTD infected vines [151]. Considering the co-occurrence of these indicators in the context of previous work emphasizes the plasticity of their presence and relationships to other indicator species, which can differ depending on the situation in which they are detected (variety, rootstock, environment, etc.).

Our study highlighted how drastically GTD composition can change depending on nursery provenance, even if considering the same rootstocks or varieties. Likewise, variation of GTDs can be acquired if plants belong to one variety or rootstock, or another. In general, we found that Carignan was the variety with the highest amount of GTD-related fungi, while Cabernet sauvignon was the variety with the lowest amount of GTDs. We do not know the implications that higher or lower abundance of GTD-related fungi may have after being planted. However, from this study, it is clear that nursery practices can have an impact on pathogen load and a potential influence on the future health of the plantation.

## Figures and Tables

**Figure 1 jof-08-00047-f001:**
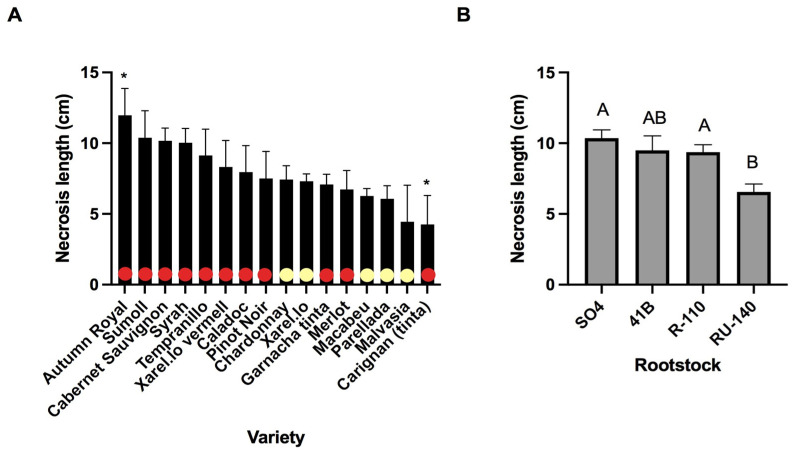
(**A**) Mean necrosis length (cm) in graft unions for each variety. Variety color is indicated with red and yellow dots for red and white varieties, respectively. Statistically significant coefficients are marked with asterisks: *p* < 0.05 (*). (**B**) Mean necrosis length in root collars from each rootstock. Post hoc comparisons using Student’s *t*-test were made and levels not connected by the same letter are significantly different.

**Figure 2 jof-08-00047-f002:**
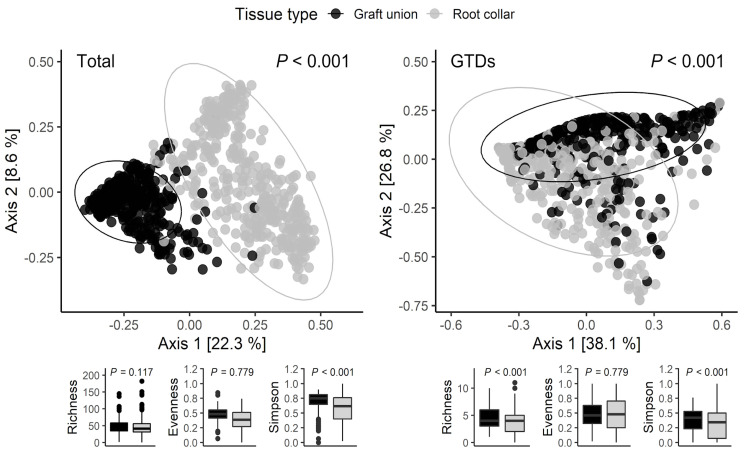
Fungal community structure comparison between all communities (**left**) and GTDs only (**right**) by graft union and root collar tissue types. The figure shows a PCoA ordination calculated with a bray-distance matrix. The R^2^ and *p*-values shown are extracted from a PERMANOVA analysis performed with the adonis2 function in R, in which all design variables are included. Alpha diversity indices (species richness, evenness, and Simpson diversity index) are presented below each figure.

**Figure 3 jof-08-00047-f003:**
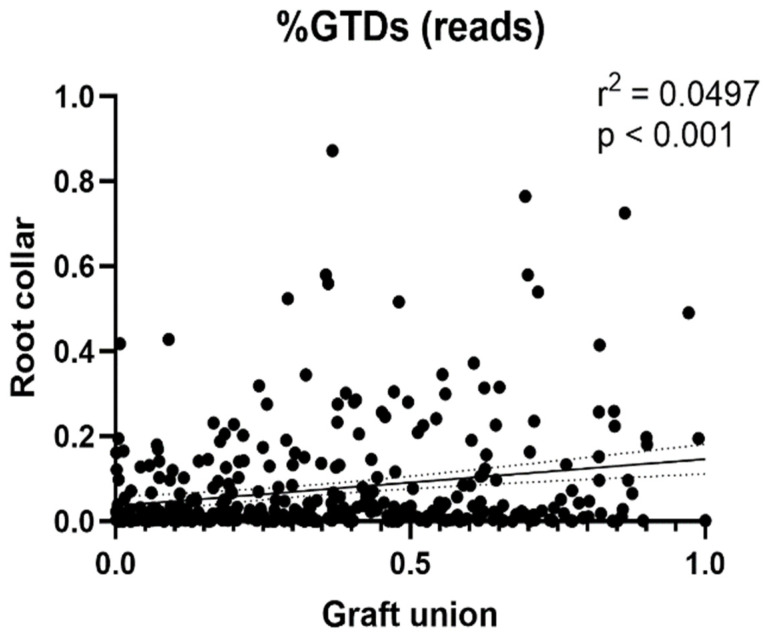
Linear regression of GTD-related OTUs (% reads) in the two tissues: correlation between graft unions and root collars.

**Figure 4 jof-08-00047-f004:**
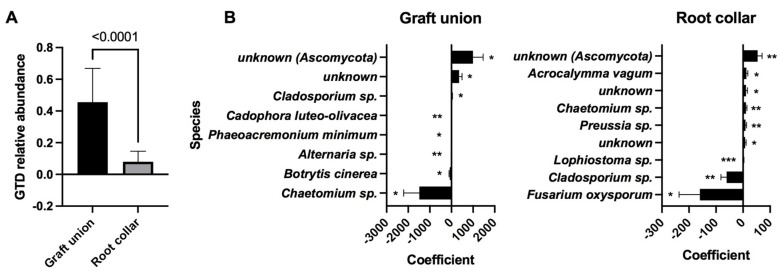
(**A**) Relative abundance (% of total reads) of GTD-related clusters in each tissue type. (**B**) Stepwise analysis of fungal communities correlated with necrosis in graft unions and root collars. Statistically significant coefficients are marked with asterisks: *p* < 0.05 (*), *p* < 0.01 (**) and *p* < 0.001 (***).

**Figure 5 jof-08-00047-f005:**
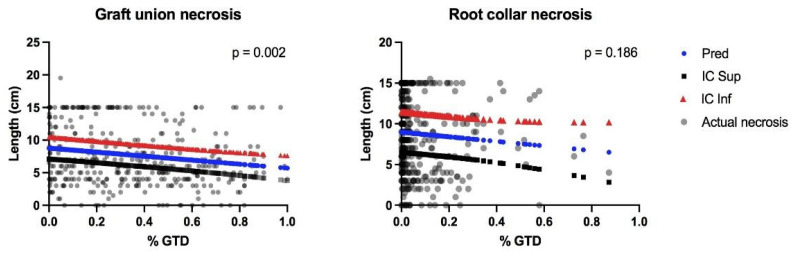
The correlation of GTD-related fungi (% GTD) to the length of necrosis (cm) in each plant tissue. Predicted (Pred) necrosis with confidence intervals (IC Sup and IC Inf) is compared with the observed (actual) necrosis.

**Figure 6 jof-08-00047-f006:**
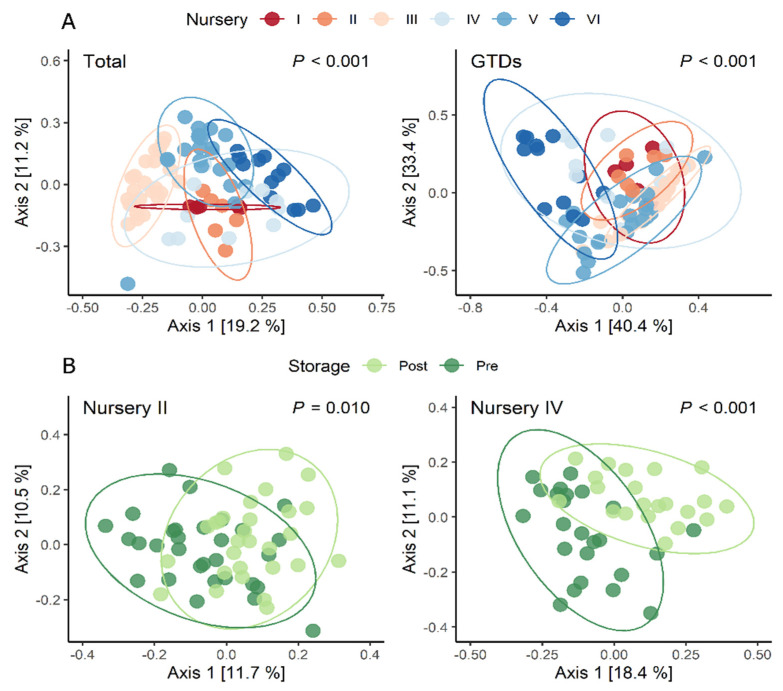
Representative examples of the degree of variation between nurseries within varieties, and of the degree of variation of storage between nurseries influencing total fungal community and GTD-related fungal community composition. (**A**) Variation in Macabeu plantings across nurseries. (**B**) Effect of storage in nurseries II and IV before (pre; dark green) and after (post; light green) storage. The figure shows a PCoA ordination calculated with a bray-distance matrix. The R^2^ and *p*-values shown are extracted from a PERMANOVA analysis performed with the adonis2 function in R, in which all design variables are included.

**Table 1 jof-08-00047-t001:** Study design. Shown are the number of plants in each scion-rootstock combination. Grafted plants were taken from six different nurseries overall. At least three nurseries per color-rootstock combination were included in replicates of six (three plants each pre- and post-storage).

Rootstock	41B	R-110	RU-140	SO4	Grand Total
*Red Varieties*		60	48	42	150
Autumn Royal		6			6
Cabernet Sauvignon		12	6	12	30
Caladoc			6		6
Carignan				6	6
Garnacha tinta		24	18		42
Merlot		6		6	12
Pinot Noir				6	6
Sumoll			6		6
Syrah		12	6	6	24
Tempranillo			6		6
Xarel.lo vermell				6	6
*White Varieties*	36	84	54	66	240
Chardonnay	6	12	6		24
Macabeu	18	36	12	24	90
Malvasia			6		6
Parellada		6	12	12	30
Xarel.lo	12	30	18	30	90
**Grand Total**	**36**	**144**	**102**	**108**	**390**

**Table 2 jof-08-00047-t002:** Contribution of nursery, variety/rootstock and storage to variation in necrosis in graft unions and root collars. Annotation next to the numbers indicates significance level: *** < 0.001, * < 0.05, n.s. = not significant (>0.05).

Factor	Graft Unions (r^2^)	Root Collars (r^2^)
Nursery	0.064 ***	0.011 n.s.
Variety	0.019 *	-
Rootstock	-	0.101 ***
Storage	0.008 n.s.	0.013 n.s.

**Table 3 jof-08-00047-t003:** Contribution of nursery, variety, color, rootstock, and storage to (**A**) total community variation in graft unions, root collars, and both tissues combined (total), compared that of (**B**) only GTDs. Annotation next to the numbers indicates significance level: *** <0.001, ** <0.01, * <0.05, n.s. = not significant (>0.05).

**A. Total Community**			
**Factor**	**Graft** **unions** **(r^2^)**	**Root collars (r^2^)**	**Total (r^2^)**
Tissue type	-	-	0.560 ***
Nursery	0.363 ***	0.159 ***	0.057 ***
Variety	0.174 ***	0.136 ***	0.045 ***
Color	0.038 ***	0.030 ***	0.012 **
Rootstock	0.047 ***	0.022 *	0.004 n.s.
Storage	0.436 n.s.	0.701 n.s.	0.760 n.s.
**B. GTD Community**			
**Factor**	**Graft unions (r^2^)**	**Root collars (r^2^)**	**Total (r^2^)**
Tissue type	-	-	0.107 ***
Nursery	0.240 ***	0.208 ***	0.179 ***
Variety	0.115 ***	0.095 ***	0.072 ***
Color	0.024 ***	0.014 ***	0.016 ***
Rootstock	0.045 ***	0.005 n.s.	0.014 **
Storage	0.003 n.s.	0.049 ***	0.016 ***

**Table 4 jof-08-00047-t004:** Contribution of nursery and storage to the composition of fungal communities including all OTUs (total) and of only OTUs related to GTDs in different varieties and rootstocks. The number in parentheses indicates the number of nurseries the variety or rootstock was present in. Only varieties that were present in at least three nurseries are shown. Annotation next to the numbers indicates significance level: *** <0.001, ** <0.01, * <0.05, n.s. = not significant (>0.05).

*Graft Union*	Nursery (r^2^)		Storage (r^2^)	
Variety	Total	GTDs	Total	GTDs
Cabernet (3)	0.490 ***	0.266 **	0.185**	0.010 n.s.
Chardonnay (4)	0.649 ***	0.346 *	0.137 n.s.	0.064 n.s.
Garnacha tinta (5)	0.457 ***	0.220 *	0.117 **	0.037 n.s.
Macabeu (6)	0.599 ***	0.448 ***	0.003 n.s.	0.000 n.s.
Xarel·lo (5)	0.430 ***	0.303 ***	0.087 **	0.038 *
** *Root collar* **				
**Rootstock**	**Total**	**GTDs**	**Total**	**GTDs**
R-110 (6)	0.184 ***	0.211 ***	0.051 **	0.052 **
SO4 (6)	0.290 ***	0.329 ***	0.069 **	0.020 n.s.
RU-140 (5)	0.247 ***	0.143 **	0.090 ***	0.099 ***
41B (3)	0.443 ***	0.315 **	0.156 *	0.219 **

**Table 5 jof-08-00047-t005:** Contribution of the investigated factors to the composition of the entire fungal community (total) and of only GTDs in each nursery. The number in parentheses indicates the number of varieties produced by each nursery. Annotation next to the numbers indicates significance level: *** < 0.001, ** < 0.01, * < 0.05, n.s. = not significant (>0.05).

*Graft union*	Variety (r^2^)		Storage (r^2^)		Rootstock (r^2^)	
Nursery	Total	GTDs	Total	GTDs	Total	GTDs
I (10)	0.337 ***	0.266 **	0.153 ***	0.021 n.s.	-	-
II (7)	0.211 *	0.270 **	0.123 ***	0.005 n.s.	-	-
III (7)	0.173 **	0.051 n.s.	0.160 ***	0.048 *	-	-
IV (4)	0.259 **	0.181 *	0.301 ***	0.106 **	-	-
V (5)	0.149 *	0.138 *	0.179 ***	0.000 n.s.	-	-
VI (3)	0.303 **	0.338 **	0.448 ***	0.316 ***	-	-
** *Root collar* **						
**Nursery**	**Total**	**GTDs**	**Total**	**GTDs**	**Total**	**GTDs**
I (10)	-	-	0.028 n.s.	0.143 ***	0.126 **	0.053 n.s.
II (7)	-	-	0.198 ***	0.005 n.s.	0.080 n.s.	0.049 n.s.
III (7)	-	-	0.033 *	0.424 ***	0.108 **	0.031 n.s.
IV (4)	-	-	0.279 ***	0.038 n.s.	0.060 n.s.	0.038 n.s.
V (5)	-	-	0.104 **	0.002 n.s.	0.110 *	0.066 n.s.
VI (3)	-	-	0.269 ***	0.133 n.s.	0.242 *	0.145 n.s.

**Table 6 jof-08-00047-t006:** Contribution of each factor to the presence of GTD-related fungal isolates in each tissue type (graft union and root collar). ChiSq (X^2^) values are reported. Annotation next to the numbers indicates significance level: *** <0.001, * <0.05, n.s. = not significant (>0.05).

Factor	Graft Union (X^2^)	Root Collar (X^2^)
Nursery	29.608 ***	24.954 ***
Variety	28.641 *	-
Rootstock	-	2.419 n.s.

## Data Availability

The data presented in this study are openly available in FigShare at https://doi.org/10.6084/m9.figshare.17089295 (accessed on 29 November 2021).

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
