# Peer review of "Variation in Fungal Community in Grapevine (*Vitis vinifera*) Nursery Stock Depends on Nursery, Variety and Rootstock"

_jof, 2022, doi:10.3390/jof8010047_

Round 1

Reviewer 1 Report

Dear Author,

Please see comments in attached file. 

Minor corrections requires through out the manuscript.

Introduction nee to be revised and precise.

Scientific names of fungi correct as mark in attached file.

Conclusion also need revision for the better understanding of readers 

Reference format check as per journal.

Reviewer 2 Report

The authors carried out an analysis of the fungal community associated with different tissues of the grapevine, considering various factors such as variety, rootstock, nursery etc. The main objective was to evaluate the influence of these variables on the presence of GTD-related species. Both metabarcoding and isolation methods were used. The manuscript is interesting, the scope well identified and the experimental design is adequate. However, before publication, some improvement are needed, especially regarding the statistical tests used and terminology. Please see the comments below:

Page 5, section 2.3 more details about PCR condition must be included

Figure 2: a, b and c letters in the figure are missing. In the text is reported fig. 2A (uppercase) while lowercase is used in the figure, please be consistent.

Figure 2 legend: what mean “clusters”, please explain

Page 7: “the most significant variation in terms of both the fungal community and GTD related fungi occurred between graft unions and root collars” please explain better what mean “variation” between ….,  is it mean the higher difference in the level of variation ?, I suggest to replace with “difference”.

Page 7: “accounting for 17% of the variation (p < 0.01) (Figure 2A)” it is not clear to me how this value can be read from fig. 2A. Moreover, what kind of analysis is depicted in fig. 2 upper part? Is it a PCA or PCOa, or other ? in this last case what distance index was used? Please explain better and add in M&M. Also, a comment about this analysis must be added in the text.

Page 8: “Variety was also an important factor, which accounted for 17% variation in graft unions and 13% in root collars (Table 2).” Please correct, this is table 3 not table 2

Page 9: Figure 3 is presented here but it is not commented in the text (neither in the results nor in the discussion). Please comment or remove

I do not understand well how differences were evaluated for each factor (are parametric methods adequate?), in my opinion the best method should be calculating beta diversity index such as Jaccard or Bray-Curtis and performing an ordination analysis (PCoa or NMMDS), followed by a non-parametric statistical test such as PERMANOVA or ANOSIM.

Page 9: “We found the accumulation of GTD-related clusters…” what means “accumulation ……….clusters” ? Please explain better (see also previous comment). Why not simply using “the total n. of GTD-related OTUs (or reads)” instead ?

Page 10, section 3.2: “Because of that, we analyzed the effect of nurseries within varieties and the effect of varieties within nurseries (Table 4)”. This sentence must be revised, as it appear wrong to me. May be corrected as follow:  " Because of that, we analyzed the effect of nurseries and storage within varieties (Table 4)."  Moreover storage within nursery was not reported.

“Variation between nurseries, within variety is 43% to 65%” . Please specify that this is for total species (For GTDs  only is quite low).

“The variation across rootstocks was similar as it ranged from 18% to 45% between nurseries” this actually is not similar but lower. Please change  "across rootstocks" with "between rootstocks". This variation is between 18 and 44% (not 45) according to table 4.

Figure 5: again, wat type of analysis is represented by the figure? (is it a PCA? PCOa?...), moreover only fig. 5A was described/cited in the text. Therefore Fig 5b must be commented/reported in the text or removed. For example, I suggest to move in the results section 3.2 the comments reported in the legend.

Figure 6: please explain better the fig 6 in the results section and M&M

Page 13: “In the root collar, the number of indicator species represented 17% of the total 17%”. Is this sentence correct? The total root collar was above reported as 20% , not 17%.

Page 13: “two species (Phomopsis sp. and Clonostachys sp.) appearing twice as different OTUs.”  Need to be rewritten.

Page 13: “GTD-related funguses” please correct “funguses” with “fungi”!

Page 13: “In the second analysis, we focussed on the core community (OTUs with less than 30reads) (Table S5B)” This is actually the contrary, the OTUs with more than 30 reads were considered according to table S5b.*

Paragraph 3.4: indicator species analysis need to be revised and partially rewritten. It is not clear to me how indicator species have been identified, which parameters/conditions were considered to evaluate species preference/association. Please enter here or in the M&M a brief explanation of the meaning of indicator species.

To this regards, the authors in the M&M wrote that the R "multipath" function was used, however the combination of condition for which association with a given specie was searched is not reported. What means “endogenous features of the plant”?

The analysis have been performed on NGS data only or the isolates have been also considered? Please explain.

At the beginning of section 3.4 134 species were considered (are these OTU ??, please be consistent with terminology), however at the section 3.2 it is reported that only the first 100 most abundant OTUs were considered for the metabarcoding analysis. Moreover later was reported that 422 species (please rename as OTUs) have been found after removing OTUs with less than 30 reads. This sound strange to me, are data (OTU) processed differently for different analyses? Please explain better (in the M&M) how data were processed for the different analyses..

Discussion

Page 14: is the term “canes” correct ? please review.

Page 14: a large section was dedicated to discuss the symptomatic versus the asymptomatic tissues, also in the introduction it is reported that symptomatic and asymptomatic tissues were analyzed but in the results, these two factors were not considered/compared.

To this regard, I understand that isolation was performed starting from symptomatic tissues as reported in paragraph 2.3 but nothing is reported about the metabarcoding. Is it performed from the same tissues or from asymptomatic tissues? This point need to be better explained in the manuscript.

Page 15: “Our work is not the first to find interaction between Clonostachys rosea and Trichoderma sp. in plants where GTDs are present.” What kind of interaction? Interaction need to be proved experimentally! Please use a different term here, for example: co-occurrence.

Page 15: “In conclusion, our study highlighted the large variation of GTDs that can be obtained by buying the same rootstock/scion combination in one nursery or another (more than 40% of the variation).” “Likewise, variation of GTDs can be acquired if plants belong to one variety or rootstock, or another”.  Can “variation” be “acquired” or “obtained”?, please rewrite this part.

For example “In conclusion, our study highlighted how GTDs composition can change depending on nursery provenance, variety or rootstock”

Reviewer 3 Report

The studied the role of the nursery, variety and rootstock in the composition of the fungal communities associated with grapevine trunk diseases. Overall the paper is well written and acceptable for publication with minor revision.

  1. The authors stated that "The study was two-fold", but then mentioned 4 objectives. If the authors can clarify this statement in the revised manuscript.
  2. The authors also explained what they will do in the manuscript (in the last part of the Introduction). But it is not link to the objectives of the study. If the authors can link this information with the aims and objective of the study?
  3. The authors stated "The experiment was conducted on ‘healthy’ bench-grafted bare-root plants that had been propagated as one plant in the field for a year". Can the authors provide more information about the plants (e.g. where)?
  4. Can the authors provide more information (e.g., at what time point did they) about this sentence "observed and transversal sections were cut to best represent the necrosis present."
  5. In the sentence "symptomatic regions (necrotic) of the root collar and graft union. Endophyte isolation", is endophyte the correct word to use? The authors also mentioned "endophytic into a pathogenic state". 
  6. Is Fifty mg of each sample the correct concentration?
